# Applications of Large Language Models in Pathology

**DOI:** 10.3390/bioengineering11040342

**Published:** 2024-03-31

**Authors:** Jerome Cheng

**Affiliations:** Department of Pathology, University of Michigan, Ann Arbor, MI 48105, USA; jeromech@med.umich.edu

**Keywords:** large language model, generative pretrained transformer, bidirectional encoder representations from transformers, artificial intelligence, natural language processing, GPT, BERT, Mistral, Llama, Gemma, surgical pathology

## Abstract

Large language models (LLMs) are transformer-based neural networks that can provide human-like responses to questions and instructions. LLMs can generate educational material, summarize text, extract structured data from free text, create reports, write programs, and potentially assist in case sign-out. LLMs combined with vision models can assist in interpreting histopathology images. LLMs have immense potential in transforming pathology practice and education, but these models are not infallible, so any artificial intelligence generated content must be verified with reputable sources. Caution must be exercised on how these models are integrated into clinical practice, as these models can produce hallucinations and incorrect results, and an over-reliance on artificial intelligence may lead to de-skilling and automation bias. This review paper provides a brief history of LLMs and highlights several use cases for LLMs in the field of pathology.

## 1. Introduction

LLMs are based on the transformer neural network architecture introduced in the 2017 paper “Attention Is All You Need” written by Vaswani et al. [1] This paper significantly changed the landscape of natural language processing by providing the fundamental architecture used in modern large language models. Although it was not evident at the time of publication, it was soon discovered that transformers were very powerful in multiple natural language tasks that previously required different algorithms to solve. The first version of generative pretrained transformers (GPT) was released by OpenAI in June 2018 [2]. Later in the same year, Bidirectional Encoder Representations from Transformers (BERT) was released by Google researchers [3]. Since then, the progress of transformers has gone a long way, and every few months, models are being released that outperform earlier models. Initially, transformers were intended to assist only in natural language tasks, such as language translation (using sequence to sequence transformers), named entity recognition [4], and text summarization, until Dosovitskiy et al. discovered that transformers may be used for image recognition by using image embeddings with position encodings as an input for standard transformers [5]. Another breakthrough was achieved in 2020, when OpenAI released the 175 billion parameter GPT3 that performed very well in multiple metrics, showing that a very high parameter count may be key to the development of general language models [6]. Later in 2022, when ChatGPT3.5 was introduced to the public, it brought about a quick surge in popularity of LLMs due to its simple and user-friendly interface, while demonstrating impressive performance in natural language understanding, and providing seemingly intelligent answers to prompts/questions. Even experts in the field were completely astounded when ChatGPT was able to simulate human understanding and generate human-like responses at a very high level [7]. Similar to how we treat other deep neural networks as a black box, we still do not fully comprehend how a neural network trained only to predict the next word (based on probabilities) can be so powerful. It can follow instructions, explain text, summarize text, carry out meaningful conversations, write computer programs, generate reports, perform text manipulations, and is proficient at generating well-written prose in different styles. In 2023, GPT4 was released, containing over 1 trillion parameters. It was significantly better than its predecessor in multilingual capabilities, contextual understanding, and reasoning skills [8].

The earlier generation of LLMs (e.g., BERT) had fewer parameters, and therefore were trainable on relatively smaller resources (e.g., 16 TPU chips) [3]. BERT (Bert large) has 345 million parameters, which is small compared to GPT4 and many recently released LLMs. Notwithstanding, BERT is still a very useful model, and its relatively small size makes it accessible to individuals and groups with fewer resources. Several groups have been able to train LLMs from scratch. Geiping and Goldstein showed that it was possible to train a language model with similar performance to BERT by using only a single consumer GPU in a single day [9]. Mitchell et al. trained a BERT transformer model with 275,605 pathology reports, which included 121 million words, to extract data from pathology reports using eight Nvidia Tesla V100 GPUs [10]. Modern LLMs have larger training set and compute requirements. To put things into perspective, Llama 65b took Meta approximately 21 Days to train using 2048 A100 GPUs with 80 GB of RAM [11], while Llama 2 70b, which is open sourced, took Meta a total of 1,720,320 GPU hours to train using the same type of GPUs [12]. These pre-trained models are referred to as foundation models. Foundation models may be fine-tuned for specific tasks through supervised learning (e.g., using a question-and-answer dataset) with much smaller GPU resources. A human feedback loop where a person decides whether an LLM-generated response is desirable or not can further refine model performance [13].

Various groups have published studies in almost every medical and non-medical field on the use of ChatGPT and other LLMs in their respective specialties, with mixed success. This review paper will focus on the applications of large language models in the field of pathology.

## 2. Education

ChatGPT shows considerable potential in education. LLMs can assist medical educators with curriculum development, writing presentations [14], formulation of course syllabuses, developing case scenarios, crafting learning plans, and developing proposals [15]. A curriculum may be created by prompting ChatGPT to create a specific curriculum with its associated requirements. In one study, ChatGPT was asked to provide models for medical education DEI (diversity, equity, and inclusion) curriculum development in a radiology residency program. ChatGPT was subsequently asked to create a curriculum based on one of the recommended models, with follow-up prompts asking ChatGPT to create surveys, goals, topics, and implementation steps [15].

ChatGPT can provide an interactive educational experience, summarize important concepts [16], and give instant access to information [17]. It can tailor the educational experience according to an individual’s needs by providing personalized responses to questions with immediate feedback [18]. ChatGPT will provide a detailed explanation if asked to explain the management of hyperlipidemia. ChatGPT may be asked to summarize topics by prompting it to “provide a summary” of the desired topic. Alternatively, an entire article may be copied onto the chat box as part of a prompt that states “summarize the following:”. Follow-up prompts can clarify specific points, errors, and inconsistencies. Subsequent prompts can be made to create multiple choice questions based on the article summary. Given a list of signs and symptoms, it can recommend a set of differential diagnoses and engage in a brainstorming session where students can ask follow-up questions [19]. LLMs can generate and organize material in different formats to facilitate learning. By specifying “in table format” in the prompt, information about certain topics can be summarized in a table (Figure 1). It can generate outlines, multiple-choice questions with answers and explanations [20], and even simulate conversations between people about a certain topic. There is a possibility that some of the information is incorrect due to the tendency of LLMs to hallucinate [19], so LLM-generated information must be verified with other sources.

In a paper using ChatGPT to answer pathology-related questions, such as “explain why transfusion-related diseases are avoidable”, it was able to provide credible responses in most cases, achieving an overall median rating of 4.08 out 5 [21]. In a recently published study, the domain-specific knowledge of ChatGPT 3.5 in pathology was assessed to be the same level as a staff pathologist, while ChatGPT 4 exceeded that of a trained pathologist [17]. There may be instances where ChatGPT may not have information about certain topics in its knowledge base. In these cases, an article may be copied as part of the input prompt, and ChatGPT may be asked to summarize the article, or provide the important points within the text. The desired length of the summary may be specified, but there is no guarantee ChatGPT will follow it.

Not all studies pertaining to ChatGPT and education were positive. Ngo et al. [20] used ChatGPT 3.5 to generate questions for an immunology course, but it was able to generate correct questions with answers and explanations in only 32% (19 of 60) of cases. In a separate study, ChatGPT4 was tested with the 2022 American Society for Clinical Pathology resident question bank and it did not fare very well, scoring 60.42% in clinical pathology, 54.94% in anatomic pathology, and garnering an overall score of 56.98% [22]. Munoz–Zuluaga et al. tested GPT4 on 65 questions related to clinical laboratory medicine, and only 50.7% of answers were evaluated to be correct [23]. In another study, clinical chemistry faculty and trainees outperformed ChatGPT (versions 3.5 and 4) in answering 35 clinical chemistry questions [24]. Considering the poor performance of ChatGPT in answering medical questions in several studies, one should not trust everything generated by ChatGPT, and information should be cross-referenced with other reputable sources, such as textbooks.

## 3. Information Extraction

Cancer registries and research studies will benefit tremendously from automated information extraction, where manual review and encoding of pathology reports are often necessary when these reports are in free text format [25]. Choi et al. showed that LLM-assisted extraction of structured information (e.g., tumor location, surgery type, histologic grade) from pathology and ultrasound reports can lead to significant time and cost savings, compared to manual methods [26]. The accuracy of GPT 3.5 in this study was mostly in the 80s and 90s (overall accuracy of 87.7%), therefore suggesting that manual supervision is still needed. Considering the study used GPT 3.5, the numbers would probably be better if newer LLMs, such as GPT4, were used.

Information extraction from clinical notes and free-text reports is challenging using traditional methods; regular expressions [27] or multiple text matching rules have to be written, and these rules can be error-prone, especially when unstructured free text is involved. Determining the presence or absence of cancer within a report is prone to error, as looking for cases with “carcinoma” could retrieve cases of “negative for carcinoma” being phrased in different ways [28]. This is one task LLMs have proven to be good at, since they are trained to learn the meaning of words, in addition to looking for the presence or absence of a particular word. Transformers have been used for predicting CPT codes [29], naming entities in breast pathology reports [30], extracting clinical information [26], and extracting structured information from free-text pathology reports [31]. One study used retrieval-augmented generation, which enabled GPT4 to extract inclusion/exclusion criteria from free-text clinical reports, leading to increased efficiency and reduced costs [32]. Zhang et al. fine-tuned a BERT model to extract various concepts (e.g., site, degree of differentiation) from breast cancer reports, achieving an overall precision of 0.927 and recall of 0.939 [30]. Yang et al. trained a BERT model that can accurately predict the type of rejection and IFTA (interstitial fibrosis and tubular atrophy) in renal pathology reports [33]. Liu et al. developed a BERT deidentification pipeline using 2100 pathology reports, achieving a best F1-score of 0.9659 in identifying sensitive health information. It was deployed in a teaching hospital and managed to process over 8000 unstructured notes in real time [34]. Santos et al. released an open source pre-trained transformer model (PathologyBERT), which was trained on 347,173 pathology reports. It was found to be accurate in breast cancer severity classification, and may be utilized for other information extraction and classification tasks involving pathology reports [35].

Many earlier studies using transformers involve pre-training and fine-tuning of the BERT transformer architecture [35]. With modern chat-based LLMs such as ChatGPT, it is possible to extract information without model finetuning. LLMs can be prompted to identify the location of the primary cancer within a report, or determine the presence of lymphovascular invasion [36]. In using LLMs to extract text, providing a correctly phrased user prompt can have a significant impact on the quality of the output. For instance, it may be necessary to instruct the LLM “You are a pathologist. Provide a concise answer.” if a shorter output is desired [37].

## 4. Text Classification

Scientific document classification is a labor-intensive task with diverse applications [38]. Since transformers are trained to learn contextual information between words, they can be used to classify documents using learned meanings and sentence embeddings [39]. A modified version of BERT (Sentence–BERT) used siamese and triplet network structures to derive meaningful sentence embeddings that may be compared using vector similarity metrics [40]. Embeddings are useful for text similarity search, text clustering, and text classification [41]. Transformer models are not necessarily superior to CNN and hierarchical self-attention networks in text classification, where the identification of keywords and phrases can be more significant, rather than the contextual meaning of words and sentences [42].

Several studies used LLMs for the classification of pathology reports and scientific literature. One study used BERT-derived embeddings with active learning approaches to classify and cluster pathology reports according to specific diagnosis and diagnostic category [43]. Fijacko et al. performed multinomial classification of abstract titles using the ChatGPT-4 application programming interface (API), through a python function call with predefined prompts, demonstrating the effectiveness of LLM-based approaches in bibliometric analysis [44]. Using optical character recognition to convert pathology reports into a textual format, Kefeli and Tatonetti trained several BERT-based models for TNM stage and cancer type classification [45,46]. Fang and Wang used several BERT models pre-trained on scientific literature for multi-label topic classification, achieving F1-scores over 90% [47].

## 5. Report and Content Generation

Many physicians, including pathologists, spend a significant amount of time writing notes and reports. LLMs can assist in medical report writing and produce presentations, potentially leading to an increase in efficiency [48]. LLMs can help automate the process of generating pathology reports [49] and summarize case visits [50]. It can populate a template using unstructured text, extract data from different sources, and combine data into a single report [51]. Reports may be reformatted by specifying the new output format, and the language can be simplified by asking it to avoid medical terms [52].

There is ongoing debate about the appropriateness of using ChatGPT in writing research manuscripts, and many scientists disapprove of this practice [53]. ChatGPT is trained on diverse and vast amounts of internet text, but it is not aware of the specific source of data it is trained on [54]. When it is prompted to provide a reference, it will respond that it cannot cite specific studies directly. Prompted to write a paper with citations, it will write a paper where some of the references may be fictitious. One study assessed the authenticity and accuracy of references ChatGPT (version 3.5) cited in 30 medical papers it generated, and it was revealed 47% were fabricated, 46% were authentic but inaccurate, and only 7% were without errors [55]. In another study, Naik et al. used ChatGPT to create a case report on synchronous bilateral breast cancer; where the generated explanations were sensible, but several errors were present in the citations [56].

Caution should be exercised when submitting papers with LLM-generated content to publishers, as journal policies vary regarding the use of LLMs. The ethical boundaries and acceptability of using AI in writing is still being discussed [57]. Some journals allow ChatGPT to be listed as a contributor in the acknowledgements section, while others explicitly prohibit listing ChatGPT as an author [58]. However, it is unclear how journals will be able to identify submissions that are AI-generated, as tools developed for this purpose can misidentify real abstracts as AI-generated [59], and researchers themselves have trouble differentiating between AI-generated and original content [60].

## 6. Prompt Engineering

Prompt engineering is the process of designing questions and instructions in order to obtain the best response from an LLM [61]. The appropriate combination of words is crucial to eliciting the most accurate and relevant response [62], although different LLMs can have a variable response to the same prompt [63]. Some trial and error is often necessary to determine the most suitable prompt for a specific task [64]. It is recommended to be specific, provide context/examples, phrase questions differently, and assign a role to the LLM [65] (e.g., you are a pathologist). In a study that extracted symptoms from medical narratives, few-shot prompting (with examples of desired input and output) had higher sensitivity and specificity than zero-shot prompting (without examples of input/output) [66]. A zero-shot prompt is one that does not contain any training data [67]. Kojima et al. found out that adding “Let’s think step by step” to a question significantly improved the performance of LLMs in zero-shot reasoning [68]. In one study identifying the presence of metastatic cancer in discharge summaries, it was revealed that clear, concise prompts with reasoning steps significantly improved performance [69]. A study by Abdullahi et al. showed that LLMs performed better with multiple choice prompts that narrowed the search space, as opposed to open-ended prompts [70].

## 7. Programming

With the aid of LLMs, developing software for pathology-related projects can become easier. LLMs will enable pathologists with little or no programming experience to design and create programs [71]. AI has numerous potential impacts in computer programming: it can increase human productivity, automate tasks, reduce errors, document processes, and assist in bug detection [72,73]. It can also aid in data preparation and the development of pathology data visualization tools, websites, and artificial intelligence software using different languages, including Python and Matlab [74]. However, human validation is required to ensure proper performance of LLM-generated code [75]. ChatGPT can help you write a program to split a whole slide image into smaller images, simply by prompting it with “write a program for tiling a whole slide image into 224 by 224 pixel tiles”. It can also translate code from one programming language to another, which is highly valuable for someone translating legacy code to a modern language, or switching deep learning code to a different framework [76].

With a few lines of code, it is possible to generate API calls to GPT4 and other LLMs for inference. Access to the ChatGPT API is not free, so if cost or data privacy is a concern [77], many open source models are available, which are already very capable, but not as performant as GPT4, in part due to the smaller parameter size of these open models. In spite of this, many relatively simple tasks (e.g., text summarization, information extraction from templates) can be performed on open source models such as Mistral [78], Llama2, and Gemma.

## 8. Clinical Pathology

ChatGPT and other LLMs can assist with the interpretation of laboratory tests [23], but they do not have the level of knowledge necessary to replace the judgment of medical personnel [79], so caution must be exercised when using LLMs to interpret laboratory tests [24]. In an assessment involving 10 clinical microbiology case scenarios, ChatGPT’s performance was above average at best, and unsatisfactory in a few cases [80]. In another study, healthcare personnel were found to be better than ChatGPT at identifying fluid contamination of basic metabolic panel results [81]. Yang et al. evaluated ChatGPT-4 on blood smear images, and found it was able to identify 88% of normal blood cells and 54% of abnormal blood cells [79]. Kumari et al. [82] conducted a comparison of LLM performance in answering 50 hematology-related questions, where ChatGPT 3.5 achieved the highest score of 3.15 out of 4. The results were promising but indicated a need for LLM validation due to potential for inaccuracies [82].

LLMs can generate incorrect transfusion medicine recommendations. ChatGPT erroneously recommended Rho(D) immune globulin to prevent the development of anti-K antibodies during pregnancy [83]. In another study, several LLMs were given blood transfusion-related questions; Google Bard, GPT 3.5, and GPT4 achieved Receiver Operating Characteristic Curve (ROC AUC) scores of 0.65, 0.90, and 0.92, respectively [84].

## 9. Multi-Modal Large Language Models

LLMs that are trained with text and other types of data are referred to as multi-modal large language models (MLLMs) [85]. Of particular interest are recently developed MLLMs that can provide textual descriptions of microscopical images. MLLMs can also be tailored for object detection [86], and therefore may be used for counting cells and mitosis counting. Off-the-shelf MLLMs, such as GPT4v, are not well-trained on pathology images, so are not very suitable for analyzing histopathological images. In one study involving 100 colorectal polyp photomicrographs, ChatGPT achieved a sensitivity of 74% and specificity of 36% in adenoma detection [87]. Sievert et al. trained ChatGPT with 16 oropharyngeal confocal laser endomicroscopy images (8 with squamous cell carcinoma, 8 with normal mucosa) and it was tested with 139 images (83 with squamous cell carcinoma and 56 with healthy normal mucosa), achieving an overall accuracy of 71%, demonstrating an ability for few-shot learning. It was interesting that ChatGPT aborted the experiment when the terms “healthy” and “malignant” were used and recommended consulting a medical professional, so alternative terms had to be used to pursue the experiment [88]. However, it is highly likely that future versions of ChatGPT will fare better at interpreting histopathology images as more publicly available image/text pathology datasets become part of its training set.

Some groups fine-tuned MLLMs with surgical pathology images, with promising results. Tsuneki et al. combined convolutional neural network derived features with a recurrent neural network to generate histopathological descriptions of adenocarcinoma cases [89]. Their best model achieved a BLEU-4 score of 0.324. Sengupta and Brown demonstrated that whole slide image descriptions can be generated by combining encodings from a pre-trained vision transformer with a pre-trained BERT model, achieving a BLEU-4 score of 0.5818 [90]. Additionally, their best performing model predicted the tissue type with 79.98% accuracy, and patient gender with 66.36% accuracy. Sun et al. utilized 207,000 pathology image–text pairs to fine-tune a pre-trained OpenAI CLIP base model [91], and combined it with a 13B parameter LLM to develop PathAsst [92], which exhibited an impressive performance in interpreting pathology images. In another study, Yu et al. developed a vision-language pathology artificial intelligence (AI) assistant, named PathChat, by using 100 million histology images, 1.18 million pathology image–caption pairs, and 250,000 visual language instructions. Despite only using a 13B model combined with a vision transformer, PathChat [93] outperformed ChatGPT4v in interpreting pathology microscopic images, which was not totally unexpected, but this reinforces the notion that model size is not the only criteria for predicting performance. For a list of other MLLMs, Zhang et al. compiled a list of foundation models in healthcare, which includes several other MLLMs trained with pathology material [94].

Recently, there have been concerns that AI tools such as the aforementioned will replace pathologists, but the consensus is that these AI-based tools will supplement and augment the performance of pathologists and laboratory personnel, which would lead to increased efficiency and cost savings [95]. AI tools can be assigned to routine and boring tasks, like mitosis counting, cell counting, looking for metastasis in lymph nodes, or tasks that computers are inherently good at, such as calculating the tumor percentage on a digital slide. Additionally, adopting AI technologies is expected to improve the detection of rare events, diagnostic accuracy, and report quality [96].

## 10. Challenges and Limitations

Although LLMs show tremendous potential in improving the practice of pathology, LLMs can be prone to bias and knowledge plagiarism [97], and can also make mistakes. The core neural network of LLMs has no information beyond the time of its training, and most cannot access internet data, except for a few products that allow the LLM to search the internet for additional information [98]. Therefore, most LLMs will not be knowledgeable of recently updated clinical guidelines. LLMs can produce hallucinations [99] and recommend products/tools that may not exist. Retrieval augmented generation (RAG) methods offer promise in mitigating some of these deficiencies [100]. RAG comprises a set of techniques and methodologies that enable a large language model to query an external data source, potentially improving the accuracy and relevance of responses while reducing the incidence of hallucinations and inappropriate responses [101]. Building guardrails into LLMs is another way of reducing the occurrence of improper responses [102].

LLMs can automatically generate reports based on provided data but, due to the unpredictable nature of LLMs, care must be undertaken in integrating these technologies into pathology practice, to mitigate the risk of introducing errors into patient reports. LLM-generated reports may contain inaccuracies, biases, and fictitious content. One study noted that ChatGPT 3.5 produced a high rate of fabricated references, so particular attention should be given to LLM-generated references [55]. In its current state, there is a need for manual supervision to ensure no improper content is introduced into LLM-created content, and it is not advisable to incorporate LLMs into a fully automated workflow.

Some groups have voiced concerns that over-reliance on AI tools for pathology case sign-out may lead to deskilling, burnout, and diminished knowledge of the histology and mechanisms of disease [103,104]. Pathologists could end up using an AI tool to sign out a case without critically analyzing the case [105]. Automation bias is a closely related problem, which is the propensity of humans to believe that AI is right, in spite of information to the contrary [106]. Therefore, it is important for medical practitioners to be cognizant that their own judgments are valuable and be aware of the limitations of AI tools they use in practice. Inadequate understanding of how artificial intelligence models work is a key contributing factor to automation bias [106]. An AI-recommended diagnosis may convince a pathologist to alter a correct diagnosis into an incorrect diagnosis suggested by an AI-based tool [107]. Pathologists need to be informed that AI tools make predictions based on large amounts of mathematical computations, and that these predictions are prone to error. Adding a confidence score to predictions may help a pathologist focus on cases where an AI tool is “unsure” of its recommendations [107]. On the other hand, an AI tool may also associate a high confidence score with a wrong diagnosis, so it would be advisable for a pathologist to consult other experts when faced with difficult cases. Knowing how these models are trained can help clarify misconceptions about AI [108].

Implementation of LLM tools in healthcare settings are plagued by concerns over cost, computing power, and data privacy. Although GPT4 currently performs better than open source LLMs, there are circumstances where open LLMs are preferable, due to lack of adherence to HIPAA (Health Insurance Portability and Accountability Act) regulations, and other data security concerns [109] involving commercial LLMs. Smaller LLMs, such as the open source FastChat-T5 3B-parameter model can be run locally on a personal computer, while preserving patient privacy [36]. Other small LLM alternatives include TinyLLama [110], Microsoft’s Phi-2, and Google’s Gemma. Small LLMs come with their own set of challenges. These LLMs give less accurate responses, and some can, rarely, produce nonsensical output when given a certain prompt, repeating the same character over and over again.

## 11. Conclusions

The capabilities of LLMs are growing at a very fast rate, and LLMs will continue to get better. Smaller and open source LLMs are not as powerful, but these are still adequate for many natural language tasks and may be run locally within an institutional server. Eliciting the correct response from LLMs can benefit from carefully crafted prompts, which may involve some trial and error. LLMS, together with MLLMs, can transform many aspects of pathology practice, freeing up time from repetitive, routine, and boring tasks, such as cell counting, mitosis counting, report generation, information extraction, and medical coding. On a cautionary note, it is important that pathologists understand AI-based tools can make errors while sounding confident and be aware of the phenomenon of automation bias. LLMs can hallucinate, generating fictitious content and references. When in doubt, output from LLMs should be doublechecked with information from reliable sources.

## 12. Future Directions

It is perceivable that both open and proprietary LLMs/MLLMs will continue to improve. Additional guardrails and training incorporating human feedback is envisioned to improve LLM and MLLM performance, while reducing the incidence of erroneous responses [93]. Some errors produced by LLMs may be attributable to biases and errors present in its training set, so efforts will be undertaken to improve training data quality [111]. More publicly available pathology text and text/image datasets will become available, which will open up opportunities for training more accurate and powerful models. More open source models of varying parameter sizes will be released, including smaller LLMs that can function without a GPU, such as Tinyllama, and the 2B parameter Gemma. Guidelines will be developed to address the ethical aspects of LLM use in scientific writing and other aspects of daily practice [98]. Additional safeguards will be built into LLMs to curtail improper usage.

## Figures and Tables

**Figure 1 bioengineering-11-00342-f001:**
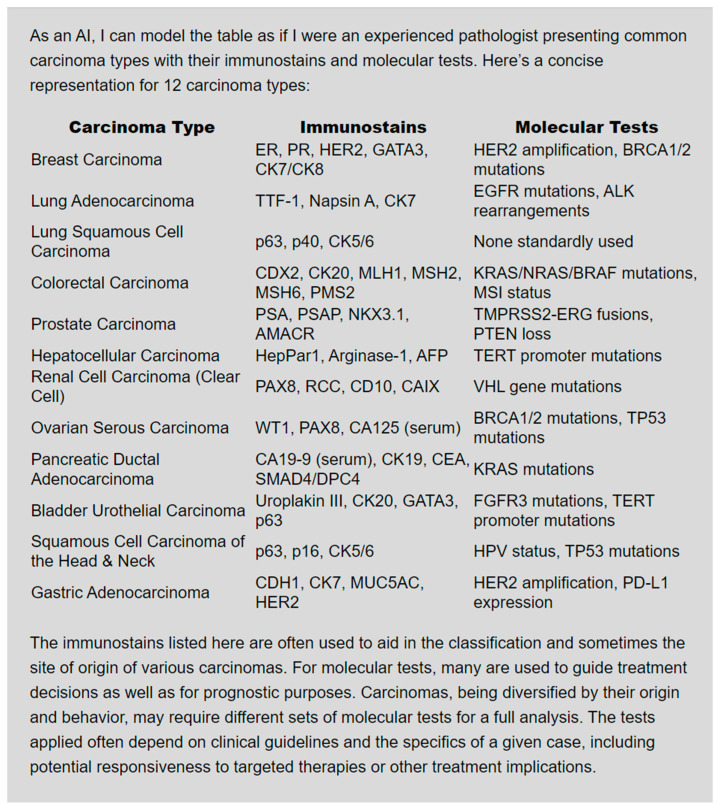
ChatGPT-4 Turbo followed instructions appropriately after being given the following prompt: “You are an experienced pathologist. Give me a list of 12 carcinomas with associated immunostains and molecular tests in table format”.

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
