# Peer review of "Applications of Large Language Models in Pathology"

_bioengineering, 2024, doi:10.3390/bioengineering11040342_

Round 1
Reviewer 1 Report
Comments and Suggestions for Authors
The manuscript "Applications of Large Language Models in Pathology" is a very interesting review that is quite informative and up-to-date. The writing style is also appreciable.
In general, the manuscript is fit to be accepted.
However, it is suggested that the author may include some relevant success stories in related other areas (viz. hematology, in doi: 10.7759/cureus.43861) in the review.
Rest, the review is worth publishing
Author Response
Thank you for the favorable review and constructive comments.
1.However, it is suggested that the author may include some relevant success stories in related other areas (viz. hematology, in doi: 10.7759/cureus.43861) in the review.
The following was added to the Clinical Pathology section:
Kumari et. al. conducted a comparison of LLM performance in answering 50 hematology related questions, where ChatGPT 3.5 achieved the highest score of 3.15 out of 4. The results were promising but indicated a need for LLM validation due to potential for inaccuracies.
The following were added to the Information Extraction section:
Zhang et. al. fine-tuned a BERT model to extract various concepts (e.g. site, degree of differentiation) from breast cancer reports, achieving an overall precision of 0.927 and recall of 0.939 [30]. Yang et. al. trained a BERT model that can accurately predict the type of rejection and IFTA (interstitial fibrosis and tubular atrophy) in renal pathology reports [33]. Liu et. al. developed a BERT deidentification pipeline using 2100 pathology reports, achieving a best F1-score of 0.9659 in identifying sensitive health information. It was deployed in a teaching hospital, and managed to process over 8000 unstructured notes in real time [34]. Thiago et. al. released an open-source pre-trained transformer model (PathologyBERT), which was trained on 347,173 pathology reports. It was found to be accurate in breast cancer severity classification, and may be utilized for other information extraction and classification tasks involving pathology reports [35].
Reviewer 2 Report
Comments and Suggestions for Authors
To enhance the paper, several key strategies can be implemented:
- Incorporate More Case Studies: Including more case studies or real-world examples that showcase the practical application of Large Language Models (LLMs) in pathology would provide concrete evidence of their effectiveness
- Discuss Ethical Considerations: It would be beneficial to delve deeper into the ethical considerations surrounding the use of LLMs in pathology practice, such as the need for verification from reputable sources and the potential risks of incorrect results and automation bias
- Explore Future Directions: Incorporating more insights into future research directions and development in the field of LLMs in pathology would offer a comprehensive overview of the potential advancements and challenges that lie ahead
- Provide Concrete Examples in Education: Further elaborating on how LLMs can be effectively utilized in educational settings, such as assisting in curriculum development, providing interactive learning experiences, and summarizing complex concepts, would offer practical insights for educators and students
- Address Limitations and Challenges: A detailed discussion on the challenges and limitations of using LLMs in clinical pathology, including the need for manual supervision and the importance of carefully integrating LLMs into practice, would provide a balanced perspective on the technology’s implications
- Highlight Success Stories: Including success stories or positive outcomes from studies that have utilized LLMs in pathology, particularly in areas such as text classification, information extraction, and report generation, would showcase the potential benefits of integrating LLMs into practice
The paper's english is good enough.
Author Response
Thank you for the favorable review and constructive comments.
1.Incorporate More Case Studies: Including more case studies or real-world examples that showcase the practical application of Large Language Models (LLMs) in pathology would provide concrete evidence of their effectiveness.
The following was added to the Clinical Pathology section:
Kumari et. al. conducted a comparison of LLM performance in answering 50 hematology related questions, where ChatGPT 3.5 achieved the highest score of 3.15 out of 4. The results were promising but indicated a need for LLM validation due to potential for inaccuracies [82].
The following were added to the Information Extraction section:
Zhang et. al. fine-tuned a BERT model to extract various concepts (e.g. site, degree of differentiation) from breast cancer reports, achieving an overall precision of 0.927 and recall of 0.939 [30]. Yang et. al. trained a BERT model that can accurately predict the type of rejection and IFTA (interstitial fibrosis and tubular atrophy) in renal pathology reports [33]. Liu et. al. developed a BERT deidentification pipeline using 2100 pathology reports, achieving a best F1-score of 0.9659 in identifying sensitive health information. It was deployed in a teaching hospital and managed to process over 8000 unstructured notes in real time [34]. Thiago et. al. released an open-source pre-trained transformer model (PathologyBERT), which was trained on 347,173 pathology reports. It was found to be accurate in breast cancer severity classification and may be utilized for other information extraction and classification tasks involving pathology reports [35].
2.Discuss Ethical Considerations: It would be beneficial to delve deeper into the ethical considerations surrounding the use of LLMs in pathology practice, such as the need for verification from reputable sources and the potential risks of incorrect results and automation bias
The following were added to the Challenges and Limitations section:
LLMs can automatically generate reports based on provided data, but due to the unpredictable nature of LLMs, care must be undertaken in integrating these technologies into pathology practice to mitigate the risk of introducing errors into patient reports. LLM generated reports may contain inaccuracies, biases, and fictitious content. One study noted that ChatGPT 3.5 produced a high rate of fabricated references, so particular attention should be placed on LLM generated references [55]. In its current state, there is a need for manual supervision to ensure no improper content is introduced into LLM created content, and it is not advisable to incorporate LLMs into a fully automated workflow...Inadequate understanding of how artificial intelligence models work is a key contributing factor to automation bias [106]. An AI recommended diagnosis may convince a pathologist to alter a correct diagnosis into an incorrect diagnosis suggested by an AI-based tool [107]. Pathologists need to be informed that AI tools make predictions based on large amounts of mathematical computations, and that these predictions are prone to error. Adding a confidence score to predictions may help a pathologist focus on cases where an AI tool is “unsure” of its recommendations [107]. On the other hand, an AI tool may also associate a high confidence score to a wrong diagnosis, so it would be advisable for a pathologist to consult other experts when faced with difficult cases...Although GPT4 currently performs better than open-source LLMs, there are circumstances where open LLMs are preferable, due to lack of adherence to HIPAA (Health Insurance Portability and Accountability Act) regulations, and other data security concerns [109] involving commercial LLMs.
3.Explore Future Directions: Incorporating more insights into future research directions and development in the field of LLMs in pathology would offer a comprehensive overview of the potential advancements and challenges that lie ahead
The following were added to the Future Directions section:
Additional guardrails and training incorporating human feedback is envisioned to improve LLM and MLLM performance, while reducing the incidence of erroneous responses [93]. Some errors produced by LLMs may be attributable to biases and errors present in its training set, so efforts will be undertaken to improve training data quality [111]. More Publicly available pathology text and text/image datasets will become available, which will open up opportunities for training more accurate and powerful models... Guidelines will be developed to address the ethical aspects of LLM use in scientific writing and other aspects of daily practice [98].
4.Provide Concrete Examples in Education: Further elaborating on how LLMs can be effectively utilized in educational settings, such as assisting in curriculum development, providing interactive learning experiences, and summarizing complex concepts, would offer practical insights for educators and students
The following were added to the Education section:
A curriculum may be created by prompting ChatGPT to create a specific curriculum with its associated requirements. In one study, ChatGPT was asked to provide models for medical education DEI (Diversity, Equity, and Inclusion) curriculum development in a Radiology Residency Program. ChatGPT was subsequently asked to create a curriculum based on one of the recommended models, with follow-up prompts asking ChatGPT to create surveys, goals, topics, and implementation steps [15] ...ChatGPT will provide a detailed explanation if asked to explain the management of hyperlipidemia. ChatGPT may be asked to summarize topics by prompting it to "provide a summary" of the desired topic. Alternatively, an entire article may be copied onto the chat box as part of a prompt that states "summarize the following:". Follow-up prompts can clarify specific points, errors, and inconsistencies. Subsequent prompts can be made to create multiple choice questions based on the article summary...There is a possibility that some of the information is incorrect due to the tendency of LLMs to hallucinate [19], so LLM generated information must be verified with other sources.
5.Address Limitations and Challenges: A detailed discussion on the challenges and limitations of using LLMs in clinical pathology, including the need for manual supervision and the importance of carefully integrating LLMs into practice, would provide a balanced perspective on the technology’s implications
The following were added to the Challenges and Limitations section:
LLMs can automatically generate reports based on provided data, but due to the unpredictable nature of LLMs, care must be undertaken in integrating these technologies into pathology practice to mitigate the risk of introducing errors into patient reports. LLM generated reports may contain inaccuracies, biases, and fictitious content. One study noted that ChatGPT 3.5 produced a high rate of fabricated references, so particular attention should be placed on LLM generated references [55]. In its current state, there is a need for manual supervision to ensure no improper content is introduced into LLM created content, and it is not advisable to incorporate LLMs into a fully automated workflow...Inadequate understanding of how artificial intelligence models work is a key contributing factor to automation bias [106]. An AI recommended diagnosis may convince a pathologist to alter a correct diagnosis into an incorrect diagnosis suggested by an AI-based tool [107]. Pathologists need to be informed that AI tools make predictions based on large amounts of mathematical computations, and that these predictions are prone to error. Adding a confidence score to predictions may help a pathologist focus on cases where an AI tool is “unsure” of its recommendations [107]. On the other hand, an AI tool may also associate a high confidence score to a wrong diagnosis, so it would be advisable for a pathologist to consult other experts when faced with difficult cases...Although GPT4 currently performs better than open-source LLMs, there are circumstances where open LLMs are preferable, due to lack of adherence to HIPAA (Health Insurance Portability and Accountability Act) regulations, and other data security concerns [109] involving commercial LLMs.
6.Highlight Success Stories: Including success stories or positive outcomes from studies that have utilized LLMs in pathology, particularly in areas such as text classification, information extraction, and report generation, would showcase the potential benefits of integrating LLMs into practice
The following were added to the Information Extraction section (articles also included in response to comment #1):
Zhang et. al. fine-tuned a BERT model to extract various concepts (e.g. site, degree of differentiation) from breast cancer reports, achieving an overall precision of 0.927 and recall of 0.939 [30]. Yang et. al. trained a BERT model that can accurately predict the type of rejection and IFTA (interstitial fibrosis and tubular atrophy) in renal pathology reports [33]. Liu et. al. developed a BERT deidentification pipeline using 2100 pathology reports, achieving a best F1-score of 0.9659 in identifying sensitive health information. It was deployed in a teaching hospital and managed to process over 8000 unstructured notes in real time [34]. Thiago et. al. released an open-source pre-trained transformer model (PathologyBERT), which was trained on 347,173 pathology reports. It was found to be accurate in breast cancer severity classification and may be utilized for other information extraction and classification tasks involving pathology reports [35].